# Chrysosplenol D Triggers Apoptosis through Heme Oxygenase-1 and Mitogen-Activated Protein Kinase Signaling in Oral Squamous Cell Carcinoma

**DOI:** 10.3390/cancers13174327

**Published:** 2021-08-27

**Authors:** Ming-Ju Hsieh, Chia-Chieh Lin, Yu-Sheng Lo, Yi-Ching Chuang, Hsin-Yu Ho, Mu-Kuan Chen

**Affiliations:** 1Oral Cancer Research Center, Changhua Christian Hospital, Changhua 500, Taiwan; 170780@cch.org.tw (M.-J.H.); 181327@cch.org.tw (C.-C.L.); 165304@cch.org.tw (Y.-S.L.); 177267@cch.org.tw (Y.-C.C.); 2College of Medicine, National Chung Hsing University, Taichung 402, Taiwan; 3Institute of Medicine, Chung Shan Medical University, Taichung 402, Taiwan; 4Graduate Institute of Biomedical Sciences, China Medical University, Taichung 404, Taiwan; 5Department of Otorhinolaryngology, Head and Neck Surgery, Changhua Christian Hospital, Changhua 500, Taiwan

**Keywords:** chrysosplenol D, apoptosis, autophagy, heme oxygenase-1, oral squamous cell carcinoma

## Abstract

**Simple Summary:**

Oral squamous cell carcinoma (OSCC) accounts for the most malignancies. A GLO-BOCAN 2020 report estimated 377,713 new cases of oral cancer and 177,757 deaths due to oral cancer in 2020. Chrysosplenol D, a flavonol isolated from *Artemisia annua* L., can exert an-ticancer effects. This study investigated the anticancer property of chrysosplenol D and its un-derlying mechanism in oral squamous cell carcinoma. We observed that chrysosplenol D reduced cell viability, cell cycle arrest, apoptosis and autophagy in OSCC. Moreover, the upregulation of heme oxygenase-1 (HO-1) was found to be critical for chrysosplenol D-induced apoptotic cell death that patients with head and neck cancer had lower HO-1 expression. The findings of the present study indicated that chrysosplenol D exerts anticancer effects on OSCC by suppressing the MAPK pathway and activating HO-1 expression. Suggest that chrysosplenol D might be a potential anticancer agent for treating OSCC.

**Abstract:**

Chrysosplenol D, a flavonol isolated from *Artemisia annua* L., can exert anticancer effects. This study investigated the anticancer property of chrysosplenol D and its underlying mechanism in oral squamous cell carcinoma (OSCC). We observed that chrysosplenol D reduced cell viability and caused cell cycle arrest in the G_2_/M phase. The findings of annexin V/propidium iodide staining, chromatin condensation, and apoptotic-related protein expression revealed that chrysosplenol D regulated apoptosis in OSCC. Furthermore, chrysosplenol D altered the expression of the autophagy marker LC3 and other autophagy-related proteins. Phosphatidylinositol 3-kinase/protein kinase B, extracellular signal-regulated kinase, c-Jun N-terminal kinase, and p38 mitogen-activated protein kinase (MAPK) were downregulated by chrysosplenol D, and the inhibition of these pathways significantly enhanced chrysosplenol D-induced cleaved poly (ADP-ribose) polymerase activation. Moreover, the upregulation of heme oxygenase-1 (HO-1) was found to be critical for chrysosplenol D-induced apoptotic cell death. The analysis of clinical data from The Cancer Genome Atlas and Gene Expression Omnibus datasets revealed that patients with head and neck cancer had lower HO-1 expression than did those with no head and neck cancer. The findings of the present study indicated that chrysosplenol D exerts anticancer effects on OSCC by suppressing the MAPK pathway and activating HO-1 expression.

## 1. Introduction

Among head and neck cancers, oral cancer results in higher morbidity than other head and neck cancers [1]. Oral cancer may originate in any part of the oral tissue and has different histological types. Oral squamous cell carcinoma (OSCC) accounts for the most malignancies. A GLOBOCAN 2020 report estimated 377,713 new cases of oral cancer and 177,757 deaths due to oral cancer in 2020 [2]. The incidence of oral cancer is high in Melanesia and South-Central Asia owing to the habit of betel nut chewing in these areas [3]. Furthermore, the incidence of oral cancer increased with age and was higher in men than in women in most regions [4]. Surgery remains the mainstay treatment for OSCC, and adjuvant therapies, such as chemoradiotherapy, epidermal growth factor receptor and cyclooxygenase-2 inhibitors, and photodynamic therapy, are used for patients with advanced disease [5,6]. Although the survival rate of patients with OSCC has been improving year by year, their prognosis is relatively poor with a high recurrence rate [7]. With the increasing incidence rates of all cancers, 28.4 million new cancer cases are predicted to occur in 2040, indicating a 47% increase in the number of cancer cases from 19.3 million new cancer cases reported in 2020 [2]. Therefore, new anticancer agents should be urgently developed.

Autophagy and apoptosis are distinct self-destructive processes that determine the turnover of cytoplasmic organelles and proteins. Autophagy involves the fusion of double-membrane vesicles, such as autophagosomes, with lysosomes to form autolysosomes, and the degradation of autophagic cargo occurs in autolysosomes [8]. Autophagy is a cellular process that protects cells against low-nutrient conditions through the participation of autophagy-related proteins [9]. Apoptosis is a type of programmed cell death and involves a series of morphological changes such as chromatin condensation, fragmentation, and apoptotic body formation [10,11]. In addition, apoptosis is characterized by biochemical changes, including apoptosis-related protein regulation (e.g., Bcl-2 family proteins), mitochondrial outer membrane permeabilization, and cysteine-aspartic protease (caspase) activation [12,13]. These two catabolic pathways are essential for organismal homeostasis and thus protect organisms against cancer [14,15]. The autophagic process mainly removes damaged organelles and proteins from cells. Tumor cells have autophagic defects that might inhibit autophagic cell death [16,17,18]. However, starvation-mediated autophagy might promote tumor survival by degrading cellular building blocks to provide nutrients for tumor cells under hypoxic and nutrient-limiting conditions [19,20]. Hence, autophagy appears to be a double-edged sword in cancer. Apoptosis is considered a tumor suppressor pathway. The disruption of both extrinsic and intrinsic pathways has been observed in several cancers. For instance, in the extrinsic pathway, the disruption often occurs due to changes in the localization of death receptors and the downregulation of the surface expression of death receptors [21,22]. In the intrinsic pathway, the disruption often occurs due to the overexpression of antiapoptotic proteins and the inactivation of proapoptotic proteins [21].

The heme oxygenase-1 (*HMOX-1*) gene encodes an essential enzyme HO-1 that is involved in heme catabolism and cleaves heme to form biliverdin, carbon monoxide, and ferrous iron. HO-1 is expressed in various types of cancers and is positively correlated with poor prognosis in patients with cancers [23]. HO-1 overexpression in cancer cells promotes proliferation, invasion, and survival [24,25,26]. However, recent studies have indicated the antitumor effects of HO-1. Gandini et al. found that HO-1 activation induced apoptosis and inhibited migration and invasion by modulating epithelial–mesenchymal transition (EMT) in breast cancer cells [27]. In addition, Yanagawa et al. reported that low HO-1 expression is correlated with an increased risk of lymph node metastasis in OSCC [28]. These findings indicate the dual role of HO-1 in cancer progression. However, the role of HO-1 in OSCC still needs further investigation.

*Artemisia annua* L. is a Chinese traditional medicine that is widely used to treat fever. Artemisinin, a sesquiterpene lactone isolated from *A. annua* L., exerts antiparasitic and anticancer effects [29,30]. In addition, *A. annua* L. contains various phenolic compounds including phenolic acids, flavonols, and flavones [31]. Few studies have reported the anticancer and anti-inflammatory properties of chrysosplenol D, a flavonol isolated from *A. annua* L. [32,33]. Moreover, flavonols such as casticin, quercetin, and kaempferol have been reported to promote cancer cell apoptosis [34,35,36]. On the basis of these findings, we hypothesized that chrysosplenol D would inhibit cancer cell proliferation in OSCC. Thus, in this study, we investigated the effects of chrysosplenol D on OSCC and elucidated its mechanism underlying cell apoptosis. In addition, we evaluated the effect of HO-1 on chrysosplenol D-treated OSCC.

## 2. Materials and Methods

### 2.1. Chemicals and Reagents

Chrysosplenol D (purity ≥98%) was purchased from ChemFaces (Wuhan Chem Faces Biochemical Co., Ltd., Wuhan, China) and dissolved in dimethyl sulfoxide (DMSO) to obtain a solution of 100 mM. Furthermore, 3-(4, 5-imethylthiazol-2-yl)-2, 5-diphenyltetrazolium bromide (MTT), 4,6-diamidino-2-phenylindole (DAPI), protease inhibitor cocktail, and phosphatase inhibitor cocktail were purchased from Sigma Aldrich (St Louis, MO, USA). Primary antibodies were purchased from Cell Signaling Technology (Danvers, MA, USA). Secondary antibodies were purchased from Jackson ImmunoResearch (West Grove, PA, USA). The final concentration of DMSO used in all treatments was <0.1%.

### 2.2. Cell Lines and Cell Culture

Human OSCC cell lines, namely SCC-9 and HSC-3, were purchased from the American Type Culture Collection (Manassas, VA, USA). The OECM-1 cell line was purchased from Bioresource Collection and Research Center (Hsinchu, Taiwan). The HSC-3-M3 cell line, which has a high metastatic potential for lymph nodes in human OSCC, is derived from the HSC-3 cell line. The HSC-3-M3 cell line was purchased from the Japanese Collection of Research Bioresources Cell Bank (Japan). For culturing, SCC-9 cells were grown in Dulbecco’s modified Eagle’s medium (DMEM): nutrient mixture F12 medium (Gibco BRL, Grand Island, NY, USA) supplemented with 10% fetal bovine serum (FBS), 1 mM L-glutamine, 1% penicillin/streptomycin, 1.5 g/L sodium bicarbonate, and 1 mM sodium pyruvate (Sigma Aldrich). HSC-3 cells were grown in DMEM medium (Gibco BRL) supplemented with 10% FBS, 1 mM glutamine, 1% penicillin/streptomycin, and 1.5 g/L sodium bicarbonate. OECM-1 cells were grown in RPMI 1640 medium (Gibco BRL) supplemented with 10% FBS, 1% penicillin/streptomycin, 2.5 mM HEPES, and 1.5 g/L sodium bicarbonate. HSC-3-M3 cells were grown in minimal essential medium (Gibco BRL) supplemented with 10% FBS, 1 mM L-glutamine, 1% penicillin/streptomycin, and 1.5 g/L of sodium bicarbonate. All cell lines were maintained at 37 °C in a humidified atmosphere of 5% CO_2_.

### 2.3. Cell Viability Assay

Cells seeded in a 96-well plate were treated with the indicated doses of chrysosplenol D (0, 25, 50, and 100 µM) for 24, 48, and 72 h, respectively. After treatment, 5 mg/mL of MTT was added to the plate with conditioned medium for 3 h. Subsequently, formazan accumulated in cells was dissolved in DMSO, and absorbance was measured at a wavelength of 595 nm by using a microplate reader (BioTek, Winooski, VT, USA).

### 2.4. Clonogenic Assay

The clonogenic assay is a cell survival assay based on the growth of a single cell into a colony [37]. Briefly, SCC-9, HSC-3, OECM-1, and HSC-3-M3 cells were counted and seeded in 6-well plates, respectively. After cell adhesion, the indicated doses of chrysosplenol D (0, 25, 50, and 100 µM) were added to the wells. To maintain adequate nutrition, the culture medium was replaced every 3 days. After 2 weeks, colonies formed were fixed with 4% paraformaldehyde for 10 min, stained with 0.5% crystal violet for 10 min, and counted under a stereomicroscope.

### 2.5. Cell Cycle Analysis

Cells treated with the indicated doses of chrysosplenol D (0, 25, 50, and 100 µM) were collected and fixed with 70% ethanol for 24 h at −20 °C. After discarding ethanol, we incubated cells with the Muse cell cycle reagent (Merck Millipore, Burlington, MA, USA) for 30 min in the dark. Subsequently, cell cycle distribution was measured using a Muse cell analyzer flow cytometer (Merck Millipore), and data were analyzed using Muse Cell Soft V1.4.0.0 Analyzer Assays (Merck Millipore).

### 2.6. Western Blot Analysis

Western blotting is a technique used to measure the protein of interest through separation based on molecular weights [38]. Briefly, cells treated with the indicated doses of chrysosplenol D (0, 25, 50, and 100 µM) were collected and lysed using radioimmunoprecipitation assay buffer (protease and phosphatase inhibitor cocktails were added to prevent protein degradation). Appropriate amounts of proteins were separated through sodium dodecyl sulfate–polyacrylamide gel electrophoresis and then transferred onto 0.22-µm polyvinylidene fluoride membranes. Subsequently, the membranes were blocked with 5% skimmed milk in tris-buffered saline/Tween-20 buffer and incubated with primary antibodies against cyclin A; cyclin B; cyclin D3; cyclin E2; cyclin-dependent kinase (CDK)2; CDK4; CDK6; p21; p27; tumor necrosis factor receptor type 1-associated death domain (TRADD); decoy receptor 2 (DcR2); death receptor 5 (DR5); Bax; Bak; Bcl-xL; Bcl-2; cleaved poly (ADP-ribose) polymerase (PARP); cleaved caspase-3, -8, and -9; long chain 3 (LC3)-I/II; p62/SQSTM1; Beclin-1; autophagy-related gene (Atg)5–Atg12 complex; phospo- protein kinase B (AKT); total-AKT; phospo-extracellular signal-regulated kinase (ERK)1/2; total-ERK1/2; phospo-p38 mitogen-activated protein kinase (MAPK); p38 MAPK; phospo- c-Jun N-terminal kinase (JNK)1/2; JNK1/2; and β-actin (dilution ratio: 1:1000) overnight at 4℃, respectively. After discarding primary antibodies, we incubated the membranes with horseradish peroxidase-conjugated secondary antibodies. Subsequently, bound antibodies on the membranes were visualized using an enhanced chemiluminescent detection kit (Merck Millipore). The results were quantitated using ImageQuant LAS 4000 Mini (GE Healthcare Life Sciences, Boston, MA, USA).

### 2.7. Chromatin Condensation Assay

The protocol for the chromatin condensation assay has been described previously [39]. Briefly, cells treated with the indicated doses of chrysosplenol D (0, 25, 50, and 100 µM) were seeded in an 8-well glass chamber slide for 24 h. Subsequently, cells were fixed with 4% paraformaldehyde and stained with DAPI (50 mg/mL). Images were observed using the Olympus FluoView FV1200 confocal microscope (Olympus Corporation, Shinjuku, Tokyo, Japan).

### 2.8. Annexin V/Propidium Iodide Double Staining

Cells treated with the indicated doses of chrysosplenol D (0, 25, 50, and 100 µM) were collected and resuspended in phosphate-buffered saline (PBS) with 2% bovine serum albumin (BSA). Subsequently, cells were incubated with annexin V–fluorescein isothiocyanate solution and propidium iodide (PI) solution (BD Biosciences, San Jose, CA, USA) in the dark. The percentage of apoptotic cells was measured using a BD Accuri C6 Plus flow cytometer (BD Biosciences), and data were analyzed using BD CSampler Plus software (BD Biosciences).

### 2.9. Mitochondrial Membrane Potential Analysis

The detailed procedure for mitochondrial membrane potential analysis has been described previously [40]. Briefly, cells treated with chrysosplenol D (0 and 100 µM) were collected and stained with Muse MitoPotential dye. Subsequently, 7-aminoactinomycin D was added to cells for 5 min to detect cell viability. Cell signals were measured using a Muse cell analyzer flow cytometer, and data were analyzed using Muse Cell Soft V1.4.0.0 Analyzer Assays.

### 2.10. In Situ Immunofluorescence Assay

Cells at density of 4 × 10^5^/well were seeded in a 6-well plate. After chrysosplenol D treatment for 24 h, cells were fixed with 4% paraformaldehyde for 20 min and then incubated with 0.5% Triton X-100 for 10 min. After washing cells with PBS and drying the residual solvent, cells were fixed with 4% paraformaldehyde and then incubated with 5% BSA at room temperature for the blocking step. Cells were incubated with the LC3-I/II primary antibody overnight at 4 °C. The next day, cells were washed and incubated with the Alexa Fluor 488-conjugated Affinipure goat anti-rabbit immunoglobulin-G secondary antibody (Jackson Immuno Research, West Grove, PA, USA) for 1 h. At the end of incubation, cells were observed under a fluorescence microscope equipped with filters for UV and blue light at 488 nm.

### 2.11. Autophagosome Detection Assay

The detailed procedure for the detection of autophagic cells has been described previously [41,42]. Cells were seeded in an 8-well glass chamber slide, followed by treatment with chrysosplenol D (0, 25, 50, and 100 μM) for 24 h. Cells were stained using a cell meter autophagy assay kit (green fluorescence; AAT Bioquest, Sunnyvale, CA, USA). Autophagosomes were observed under an Olympus FluoView FV1200 confocal microscope (Olympus Corporation).

### 2.12. Specific Inhibitor Treatments

All specific inhibitors were purchased from ChemFaces. LY294002, a phosphatidylinositol 3-kinase (PI3K)/AKT inhibitor, and MAPK inhibitors, namely SP600125 (JNK inhibitor), U0126 (ERK inhibitor), and SB203580 (p38 inhibitor), were dissolved in DMSO as stocks, respectively. Cells were treated with either chrysosplenol D, specific inhibitors (PI3K/AKT, JNK, ERK, or p38 inhibitor), or both. For the co-treatment group, cells were treated with each inhibitor for 1 h. Subsequently, chrysosplenol D (100 µM) was added, and cells were incubated for 24 h. The final concentration of DMSO for all treatments was <0.1%.

### 2.13. RNA Interference Experiments

Human small-interfering ribonucleic acids (siRNAs) for HO-1 and scrambled siRNA were purchased from Cohesion Biosciences (London, UK). Cells were transfected with each siRNA by using the Turbofect reagent (Thermo Fisher Scientific; Waltham, MA, USA) according to the manufacturer’s instructions.

### 2.14. The Cancer Genome Atlas Database Analysis

By using head and neck squamous cell carcinoma (HNSCC) tissues, we analyzed the mRNA expression level of *HMOX1* between tumor (*n* = 520) and normal (*n* = 44) groups. Data regarding 43 paired tumor samples and normal adjacent tissue samples were obtained from The Cancer Genome Atlas (TCGA) database.

### 2.15. Gene Expression Omnibus Dataset Analysis

Expression data were extracted from the Gene Expression Omnibus (GEO) dataset (GSE3524) and analyzed using GraphPad Prism, V6.0 (GraphPad Software, Inc., CA, USA). The mRNA expression level of *HMOX1* was compared between normal and OSCC tissues.

### 2.16. Statistical Analysis

All statistical analyses were performed using GraphPad Prism, V6.0 (GraphPad Software, Inc., CA, USA). All values calculated using Student’s *t* test are presented as the mean ± standard deviation (SD) from three independent experiments. Differences were considered significant at a *p* value of <0.05.

## 3. Results

### 3.1. Chrysosplenol D Exhibits Antiproliferative Activity and Causes Cell Cycle Arrest in the G_2_/M Phase in Oral Squamous Cell Carcinoma (OSCC) Cell Lines

To investigate the anticancer activity of chrysosplenol D, we first analyzed the viability of OSCC cell lines treated with chrysosplenol D by using the MTT and colony formation assays. SCC-9, OECM-1, HSC-3, and HSC-3-M3 cells were treated with different doses of chrysosplenol D (0, 25, 50, and 100 µM) for 24, 48, and 72 h, respectively (Figure 1A). We observed that the viability of these four cell lines significantly decreased in dose- and time-dependent manners. Furthermore, the findings of the colony formation assay revealed the anti-proliferative effect of chrysosplenol D on OSCC cell lines (Figure 1B,C). We observed that the HSC-3-M3 cell line, a highly metastatic cell line derived from the HSC-3 cell line, exhibited similar sensitivity to chrysosplenol D-induced cell toxicity as did the HSC-3 cell line. Thus, we selected SCC-9, OECM-1, and HSC-3 cell lines for subsequent experiments.

Next, to elucidate mechanisms underlying chrysosplenol D-induced cell growth inhibition, we performed cell cycle analysis through flow cytometry. As shown in Figure 2A,B, in the chrysosplenol D-treated groups, cell cycle distribution was significantly increased in the G_2_/M phase but attenuated in the G_0_/G_1_ phase. In addition, cell cycle distribution in the S phase increased in a dose-dependent manner in SCC-9 cells after treatment with different doses of chrysosplenol D. The cell cycle is controlled through cyclins, a group of family proteins, by activating CDKs. We observed that the protein expressions of cyclin A, cyclin B, cyclin D3, cyclin E2, CDK2, CDK4, and CDK6 were significantly attenuated after chrysosplenol D treatment (Figure 2C,D). However, we noted an increased expression of CDK inhibitors, namely p21 and p27, in SCC-9, OECM-1, and HSC-3 cells. These results indicate that cell cycle arrest in OSCC may contribute to the anti-proliferative effect of chrysosplenol D.

### 3.2. Apoptotic Effect of Chrysosplenol D on OSCC Cell Lines

We examined the morphological and biochemical hallmarks of apoptosis to investigate the apoptotic effect of chrysosplenol D on OSCC cell lines [43]. Cells treated with different doses of chrysosplenol D were stained with DAPI and observed under a fluorescence microscope. Chromatin condensation, the morphological hallmark of apoptosis marked by increased bright blue fluorescence, was observed in chrysosplenol D-treated SCC-9, OECM-1, and HSC-3 cells (Figure 3A,B). Furthermore, the percentage of apoptotic cells was measured through annexin V and PI staining and flow cytometry. As shown in Figure 3C,D, total apoptotic OSCC cells, including early and late apoptotic cells, were increased to 3–4-fold after treatment with chrysosplenol D. Mitochondrial dysfunction is involved in the induction of apoptosis, thus increasing the depolarization of the transmembrane potential [44]. As shown in Figure 4A,B, the percentage of depolarized cells increased after treatment with a high dose of chrysosplenol D (100 μM).

To examine whether chrysosplenol D exerts the apoptotic effect through intrinsic and extrinsic pathways (the two main pathways of apoptosis), we evaluated cell lysis in the chrysosplenol D- and vehicle-treated groups by performing Western blot analysis. The expression of extrinsic pathway (death receptor pathway)-related proteins, namely TRADD, DcR2, and DR5, was significantly increased in OSCC cell lines (Figure 4C,D). Furthermore, the expression of proapoptotic (Bax and Bak) and antiapoptotic (Bcl-xL and Bcl-2) proteins involved in regulating apoptosis was increased and decreased, respectively, after chrysosplenol D treatment (Figure 5A,B). In addition, the expression of cleaved caspase-3, -8, and -9 and cleaved PARP that participate in the intrinsic pathway was increased after chrysosplenol D treatment (Figure 5C,D). These results demonstrate that chrysosplenol D might activate apoptosis through both intrinsic and extrinsic pathways.

### 3.3. Activation of Autophagy and the Mitogen-Activated Protein Kinase (MAPK) Pathway by Chrysosplenol D in OSCC

Studies have indicated that compounds that exert apoptotic effects may also activate autophagy [45,46]. Hence, we examined the autophagy-associated phenomenon and protein expression in chrysosplenol D-treated OSCC cell lines. As shown in Figure 6A, microtubule-associated protein 1A/1B-light chain 3 (hereafter referred to as LC3), which accumulates in autophagosomes and autolysosomes, was first examined through fluorescence microscopy. We observed that LC3 fluorescent puncta increased in a dose-dependent manner in chrysosplenol D-treated cells, and this increase was 2–3-fold higher than that in vehicle-treated cells (Figure 6B). Subsequently, we examined the formation of autophagosomes by using a cell meter autophagy assay kit. The fluorescence levels were increased in SCC-9, OECM-1, and HSC-3 cells (Figure 6C). Moreover, after chrysosplenol D treatment, the protein expression of LC3-I/II, Beclin-1, and Atg5–Atg12 complex increased, whereas that of p62/SQSTM1 decreased (Figure 6D,E).

The MAPK and PI3K/AKT pathways, which communicate signals from cell surface receptors to DNA present inside the nucleus, are involved in various cellular processes such as proliferation, apoptosis, stress responses, and differentiation [47]. To investigate the molecular mechanism of chrysosplenol D in OSCC cell lines, we analyzed the protein expression of MAPK proteins by performing Western blot analysis. As shown in Figure 7A, the active forms of phosphorylated AKT, p38, ERK1/2, and JNK1/2 were downregulated after chrysosplenol D treatment in SCC-9, OECM-1, and HSC-3 cell lines. Moreover, after chrysosplenol D treatment, the expression of phosphorylated AKT was the lowest in OECM-1 cells, whereas that of phosphorylated ERK1/2 was the lowest in HSC-3 cells (Figure 7B).

We used the PI3K/AKT inhibitor LY294002 and MAPK pathway inhibitors, namely SP600125 (JNK inhibitor), U0126 (ERK inhibitor), and SB203580 (p38 inhibitor), to investigate the relationship among chrysosplenol D-activated autophagy, apoptosis, and PI3K/AKT and MAPK signaling pathways. As shown in Figure 7C,E,G,I, cleaved PARP and LC3-I/II were the biomarkers of apoptosis and autophagy, respectively. Our results revealed that the inhibition of not only PI3K/AKT but also p38, ERK, and JNK in the three OSCC cell lines considerably enhanced chrysosplenol D-induced cleaved PARP activation (Figure 7D,F,H,J). However, the protein expression of LC3-I/II exhibited less or no difference between chrysosplenol D-treated cells and co-treated cells, indicating that chrysosplenol D might not regulate LC3-I/II expression through either the PI3K/AKT or MAPK pathway. The aforementioned results suggest that chrysosplenol D induced cleaved PARP-mediated apoptosis through PI3K/AKT and MAPK (ERK, JNK, and p38) signaling cascades in OSCC.

### 3.4. HO-1 Is Involved in Chrysosplenol D-Activated Apoptotic Cell Death in OSCC

Although HO-1 is regarded as a predictive biomarker for several cancers [23,48], the role of HO-1 in OSCC remains unclear. To examine whether HO-1 regulates chrysosplenol D-activated apoptosis, we first analyzed the HO-1 protein level in chrysosplenol D-treated OSCC cell lines. The expression level of HO-1 was significantly increased after chrysosplenol D treatment in SCC-9, OECM-1, and HSC-3 cells (Figure 8A,B). Subsequently, we used HO-1-specific siRNA to knock down HO-1 combined with or without treatment with a high dose of chrysosplenol D (100 µM). We observed that HO-1-specific siRNA significantly reversed chrysosplenol D-induced HO-1 protein expression (Figure 8C,D). However, the expression of chrysosplenol D-activated cleaved PARP decreased in the HO-1 siRNA-transfected group than in the control siRNA group (Figure 8E,F). In addition, the decrease in the viability of OSCC cell lines after chrysosplenol D treatment was reversed in the HO-1 siRNA-transfected group than in the control siRNA group (Figure 8G). We analyzed the gene expression of HO-1 (*HMOX-1*) in head and neck cancer tissues from the TCGA database and found a significantly lower *HMOX1* expression in tumors tissues than in normal tissues (Figure 8H). Furthermore, in 43 paired tumor samples and normal adjacent tissue samples analyzed from the TCGA database, *HMOX1* expression was lower in tumor samples than in normal adjacent tissue samples (Figure 8I). In addition, *HMOX-1* expression was higher in normal tissues than in tumor tissues in the GEO database (GSE3524; Figure 8J). Overall, these results suggest that increased HO-1 expression might be crucial for chrysosplenol D-induced apoptotic cell death, and the HO-1 expression level might be a biomarker for head and neck tumors.

## 4. Discussion

Chrysosplenol D is a flavonol isolated from *A annua* L., a widely used traditional Chinese medicine. Few studies have examined the anticancer effects of chrysosplenol D on leukemia cells and triple-negative breast cancer cells [32,49]. However, its anticancer potential and molecular mechanisms should be extensively investigated. In this study, we observed that chrysosplenol D induced apoptosis in OSCC cell lines through G_2_/M phase arrest, chromatin condensation, changes in mitochondrial membrane potential, and extrinsic/intrinsic pathway regulation. In addition, chrysosplenol D treatment induced autophagy in OSCC cell lines. Moreover, increased AKT, JNK, ERK, and p38 expression might be major signaling pathways involved in the induction of apoptosis by chrysosplenol D. In addition, increased HO-1 expression was found to be critical for chrysosplenol D-induced apoptosis.

Apoptotic induction in cancer cells has been widely applied in cancer therapy (e.g., the use of chemotherapeutic agents such as paclitaxel and doxorubicin) [50]. However, most chemotherapeutic agents exert cytotoxic effects on both cancer and normal cells, thus causing intolerable side effects in patients undergoing chemotherapy. Chrysosplenol D exhibited considerably lower cytotoxicity in peripheral blood mononuclear cells and normal breast epithelial cells than in cancer cells, indicating the selectivity of this flavonol for cancer cells [32]. This is the first study to demonstrate the anti-proliferative effect of chrysosplenol D on OSCC cell lines by performing cell viability and colony formation assays. On the basis of the findings of these assays, we further investigated the potential molecular mechanisms of this compound.

Uncontrolled proliferation is strongly correlated with cell cycle dysregulation in tumor cells [51]. The G_2_/M phase is one of the most prominent checkpoints in the cell cycle that is controlled by cyclin B/CDC2 [52]. Our findings revealed that chrysosplenol D treatment increased cell cycle distribution in the G_2_/M phase in OSCC cell lines with a decreased expression of cyclin B. This finding is in accordance with the effect of chrysosplenol D on G_2_/M cell cycle arrest in previous studies [32,49]. In addition, we observed an increased expression of the CDK inhibitors p21 and p27, the two crucial cell cycle regulators, in chrysosplenol D-treated OSCC cell lines. A previous study reported that the expression of p21 and p27 inhibits not only mammalian cell proliferation but also cyclin–CDK complexes [53,54]. These findings indicate that chrysosplenol D might regulate the cell cycle in OSCC cell lines by directly inhibiting cell cycle-related proteins and disrupting the cyclin–CDK connection through the upregulation of p21 expression. We observed decreased levels of cyclin D3, cyclin E2, CDK2, CDK4, and CDK6 in chrysosplenol D-treated cells, and this decrease was correlated with G_0_/G_1_ and S phases [55]. However, future studies should investigate the regulation of cell cycle transition.

Chromatin condensation is the most characteristic feature of apoptosis [56] and can be used to observe the apoptotic effect of anticancer compounds [45,57]. In addition, apoptosis can be examined by detecting mitochondrial membrane potential through JC-1 staining and plasma membrane integrity and permeability through annexin V/PI staining [56,58]. Our results revealed that chrysosplenol D treatment increased chromatin condensation, apoptotic cell number, and depolarized mitochondrial level in OSCC cell lines, indicating the induction of apoptosis by chrysosplenol D.

Extrinsic and intrinsic pathways are major signaling pathways that initiate intracellular apoptosis. The extrinsic pathway involves death receptor-mediated interaction, whereas the intrinsic pathway involves non-receptor-mediated stimuli. The initiation of the tumor necrosis factor (TNF)-related apoptosis-inducing ligand (TRAIL)/DR4/DR5 signaling pathway can drive adaptor proteins, namely Fas-associated death receptor and TRADD, thus recruiting and activating caspase-8 [59]. Activated caspase-8 can cleave the proapoptotic Bcl-2 family member Bid. Furthermore, truncated Bid can localize to mitochondria and interact with Bax and Bak to promote the release of cytochrome c, thus providing a mechanistic link between the intrinsic and extrinsic pathways [60]. We found that chrysosplenol D induced the expression of DR5 and DcR2 in OSCC cell lines. Decoy receptors, such as, DcR1, DcR2, and osteoprotegerin, lack the functional death domain, thus preventing the induction of apoptosis and the binding of TRAIL to DRs [61]. DcR may compete with agonistic receptors, such as DR4 and DR5, for TRAIL binding [62]. Furthermore, the expression of cleaved caspase-8, Bak, and Bax and downstream apoptotic molecules, including cleaved caspase-3 and -9 and PARP, increased after chrysosplenol D treatment. These findings indicate that the affinity of DcR2 to TRAIL may be lower than that of agonistic DR5 to active apoptotic processes in chrysosplenol D-treated OSCC cell lines.

Autophagy begins with the formation of phagophores (also called isolation membranes) that contain the lipid kinase vacuolar protein sorting 34–Beclin-1 complex on the membrane. This complex is usually inactivated by antiapoptotic proteins from the Bcl-2 family including Bcl-2 and Bcl-xL [63]. Gump and Thorburn demonstrated that apoptosis and autophagy are related through two autophagy proteins, namely p62 and Beclin-1 [64]. P62 not only acts as an autophagic degradation protein but also directly interacts with apoptotic proteins such as caspase-8, ERK, and TNF receptor-associated factor 6 [65,66]. During the formation of autophagosomes and autolysosomes, LC3 is conjugated on the membrane and, therefore, regarded as the marker of autophagic process activation [67]. During the early steps of the formation of autophagosomes, ATG5, ATG12, and ATG16L1 form a complex termed as the autophagy elongation complex (ATG5–12/16L1). This elongation complex is necessary to determine the site of LC3 on the autophagosomal membrane [68]. In accordance with the finding of a previous study [32], our results revealed that chrysosplenol D promoted the formation of autophagosomes in a dose-dependent manner and regulated autophagic proteins, namely p62, Beclin-1, LC3, and Atg5–Atg12 complex.

PI3K/AKT and MAPK signaling pathways regulate various biological processes in mammalian cells through cellular mechanisms and play critical roles in tumorigenesis [69,70]. However, the results of the co-treatment of chrysosplenol D and MAPK inhibitors indicated that chrysosplenol D might not regulate LC3 protein through these signaling pathways. Hence, the mechanism through which chrysosplenol D regulates autophagy should be investigated in future studies.

Anticancer compounds exert their effects on OSCC cell lines through the PI3K/AKT and MAPK signaling pathways. For instance, norcantharidin, a demethylated analog of cantharidin, activates apoptosis by triggering the p38 MAPK pathway in OSCC [71], as well as activates aglycone and genipin and suppresses OSCC growth through the PI3K/AKT/mTOR pathway [72]. Our previous study demonstrated that dehydrocrenatidine, a natural alkaloid, modulates OSCC cell apoptosis through both JNK and ERK MAPK signaling pathways [73]. Furthermore, chrysosplenol D could induce apoptosis through the ERK signaling pathway in triple-negative human breast cancer cells [32]; this finding is consistent with our results. Moreover, chrysosplenol D activated the expression of cleaved PARP through not only the ERK pathway but through the p38, JNK, and PI3K/AKT signaling pathway. The findings of the present study provide new insights into mechanisms through which chrysosplenol D regulates cellular signaling pathways in OSCC.

HO-1 is a stress-inducible enzyme that exerts cytoprotective effects against stress-related conditions. However, accumulating evidence has suggested that HO-1 plays a specific role in cancer progression, including in cell metastasis, angiogenesis, and proliferation [74]. The effect of HO-1 widely varies among different cancer types. For instance, in pancreatic cancer, HO-1 accelerated tumor angiogenesis, whereas low HO-1 increased tumor responsiveness to anticancer treatment [75,76]. In breast cancer, HO-1 overexpression showed a correlation with decreasing tumor volume in vivo and prolonged patient survival [27]. Furthermore, HO-1 was reported to be correlated with lymph node metastasis in patients with tongue squamous cell carcinoma [28]. HO-1 mediated cadmium-induced autophagy/apoptosis in OSCC [77]. Chien et al. found that demethoxycurcumin caused cell apoptosis by triggering HO-1 and the inhibitor of apoptosis proteins (IAPs) such as cellular IAP1 and X-linked IAP [78]. Therefore, we investigated the role of HO-1 in chrysosplenol D-mediated apoptosis in OSCC. We observed that chrysosplenol D caused HO-1 upregulation and subsequently promoted PARP-dependent apoptosis. The clinical data from the TCGA and GEO databases are consistent with our finding that the HO-1 expression level was higher in normal oral tissues or adjacent normal tissues than in OSCC tumor tissues. Although the relationship of chrysosplenol D with HO-1 and MAPK signaling pathways has been observed in OSCC, the effect of chrysosplenol D on OSCC should be thoroughly explored.

## 5. Conclusions

Our results suggest that chrysosplenol D might be a potential anticancer agent for treating OSCC.

## Figures and Tables

**Figure 1 cancers-13-04327-f001:**
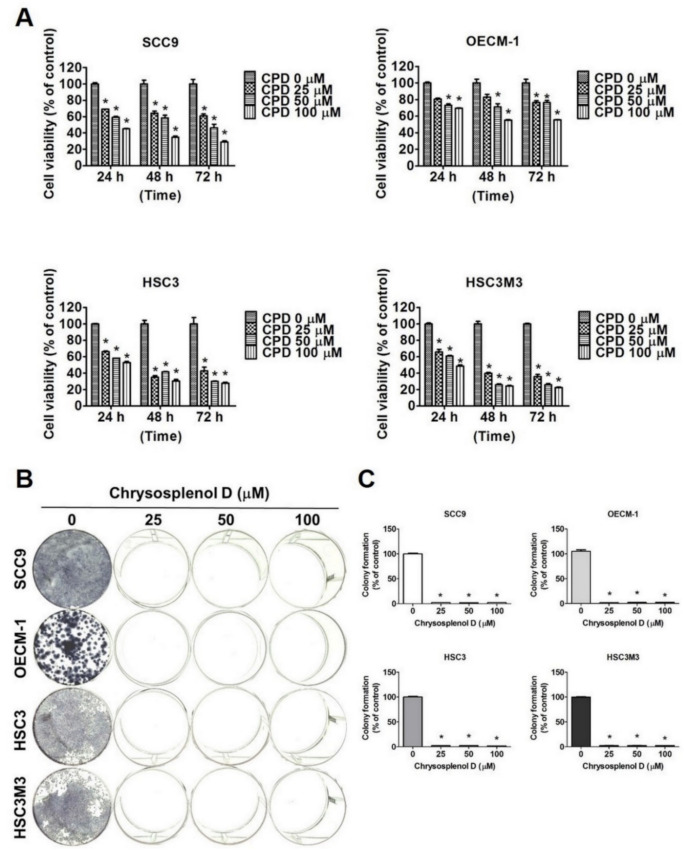
Chrysosplenol D inhibited the proliferation of oral cancer cell lines. (**A**) Human oral cancer cell lines (SCC-9, OECM-1, HSC-3, and HSC-3-M3) were treated with the indicated doses of chrysosplenol D (0, 25, 50, and 100 µM) for 24, 48, and 72 h, respectively. Cell viability was measured using the MTT assay (**B**,**C**) Cell lines were incubated with the indicated doses of chrysosplenol D (0, 25, 50, and 100 µM) for 14 days, and the culture medium was replaced every 3 days. Graphs show the findings of statistical analysis. Data are presented as the mean ± SD from three independent experiments * *p* < 0.05 compared with the vehicle treatment group.

**Figure 2 cancers-13-04327-f002:**
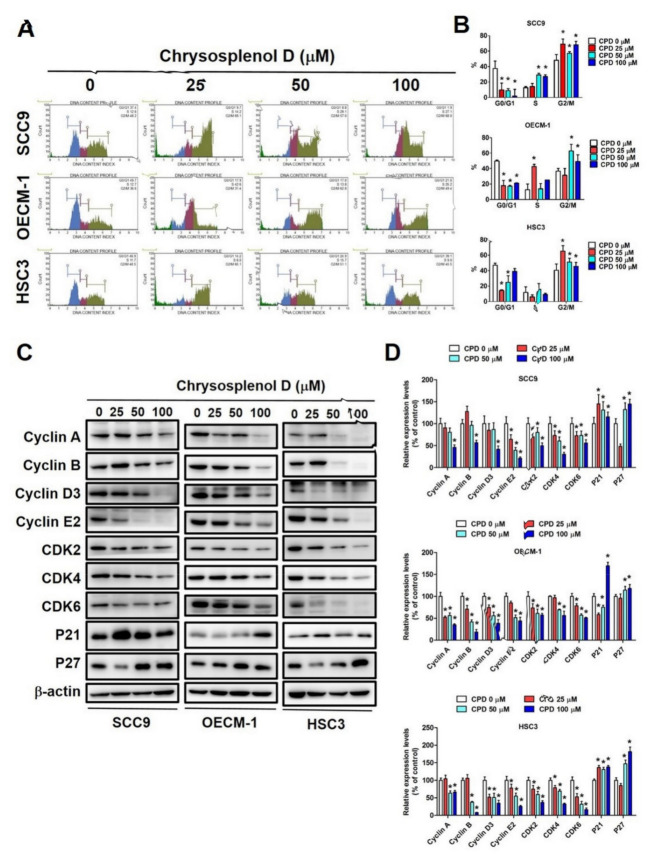
Chrysosplenol D induced cell cycle arrest in the G_2_/M phase in oral squamous cell carcinoma (OSCC) cell lines. (**A**,**B**) After the treatment of SCC-9, OECM-1, and HSC-3 cells with chrysosplenol (0, 25, 50, and 100 µM) for 24 h, cell cycle phase distribution was analyzed through flow cytometry after propidium iodide (PI) staining. Graphs show the findings of statistical analysis. (**C**,**D**) The expression levels of cell cycle-regulating proteins, namely cyclin A, cyclin B, cyclin D3, cyclin E2, CDK2, CDK4, CDK6, p21 and p27, were detected through Western blot analysis after chrysosplenol D treatment for 24 h. The β-actin protein level was used to adjust quantitative results. Graphs show the findings of the statistical analysis of cell cycle-regulating proteins. Data are presented as the mean ± SD from three independent experiments * *p* < 0.05 compared with the vehicle treatment group.

**Figure 3 cancers-13-04327-f003:**
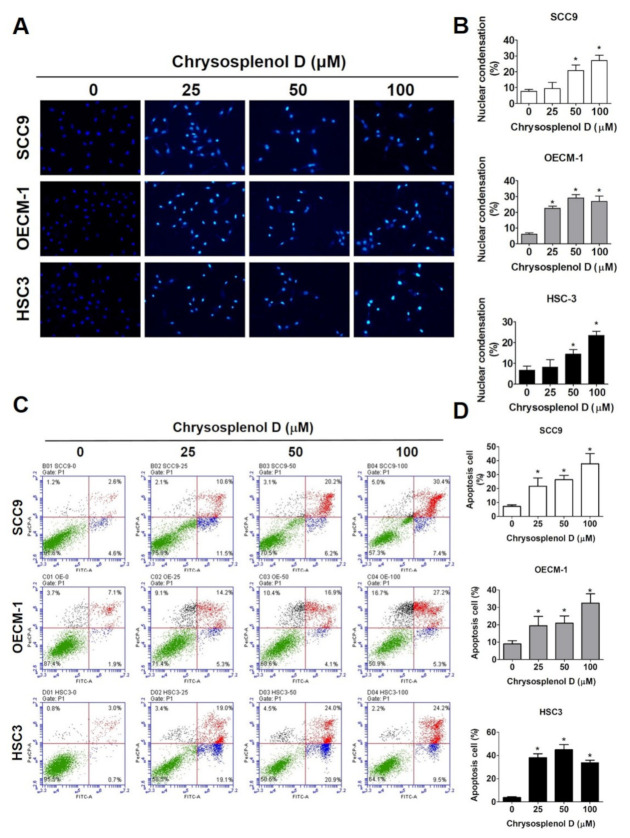
Chrysosplenol D induced apoptotic cell death in OSCC cell lines. (**A**,**B**) The morphological characteristics of apoptosis were analyzed through fluorescence microscopy after 4,6-diamidino-2-phenylindole (DAPI) staining in chrysosplenol D-treated OSCC cell lines. The bright-blue spots indicated chromatin condensation, which served as an apoptotic indicator. Graphs show the findings of the statistical analysis of chromatin condensation. (**C**,**D**) After the treatment of SCC-9, OECM-1, and HSC-3 cells with chrysosplenol D for 24 h, cell apoptosis was detected through annexin V and PI double staining and flow cytometry. Graphs show the findings of the statistical analysis of annexin V/PI staining and flow cytometry. Data are presented as the mean ± SD from three independent experiments * *p* < 0.05 compared with the vehicle treatment group.

**Figure 4 cancers-13-04327-f004:**
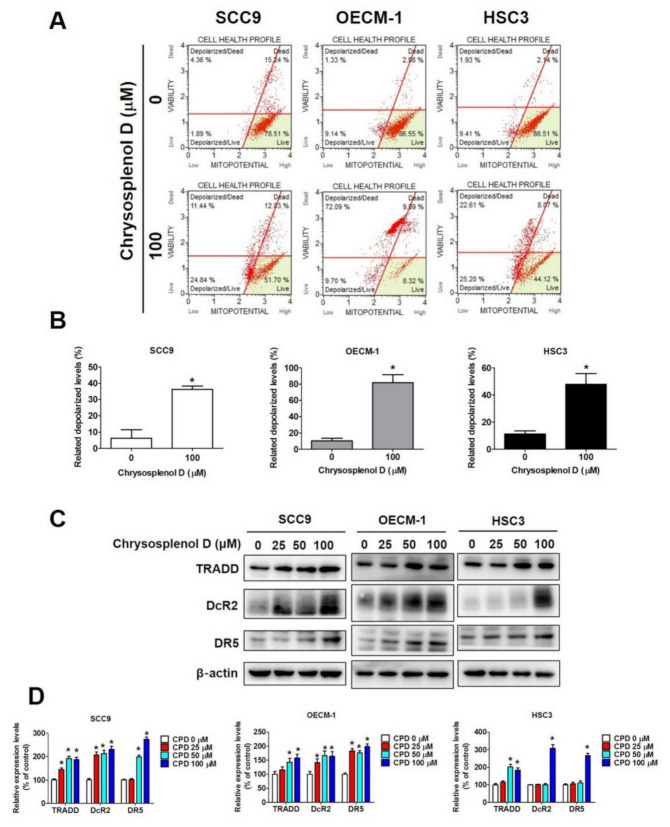
Chrysosplenol D regulated the mitochondrial membrane potential and death pathway in OSCC cell lines**.** (**A**,**B**) After chrysosplenol D (100 µM) treatment for 24 h, OSCC cells were collected and measured using a Muse cell analyzer. Quantitative data were analyzed using Muse cell software V1.4.0.0. (**C**,**D**) The expression level of death receptor proteins, namely TRADD, DcR2, and DR5, was measured through Western blot analysis. The β-actin protein level was used to adjust quantitative results. Data are presented as the mean ± SD from three independent experiments * *p* < 0.05 compared with the vehicle treatment group.

**Figure 5 cancers-13-04327-f005:**
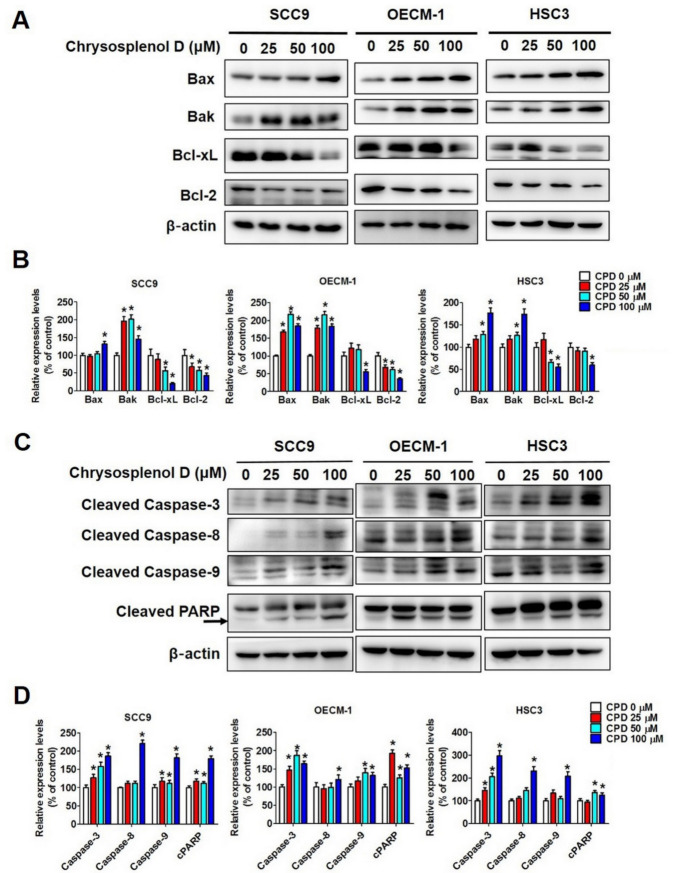
Chrysosplenol D induced apoptosis by regulating apoptotic-related proteins in OSCC cell lines**.** (**A**,**B**) Proapoptotic (Bax and Bak) and antiapoptotic proteins (Bcl-2 and Bcl-xL) were measured after chrysosplenol D treatment. The β-actin protein level was used to adjust the quantitative results of pro- and anti-apoptotic protein levels. Graphs show the findings of the statistical analysis of pro- and anti-apoptotic proteins. (**C**,**D**) The active forms of proteases (caspase-3, -8, and -9) and poly (ADP-ribose) polymerase (PARP) in apoptotic regulation were measured through Western blot analysis. The β-actin protein level was used to adjust quantitative results. Graphs show the findings of the statistical analysis of apoptotic-regulating proteins. Data are presented as the mean ± SD from three independent experiments * *p* < 0.05 compared with the vehicle treatment group.

**Figure 6 cancers-13-04327-f006:**
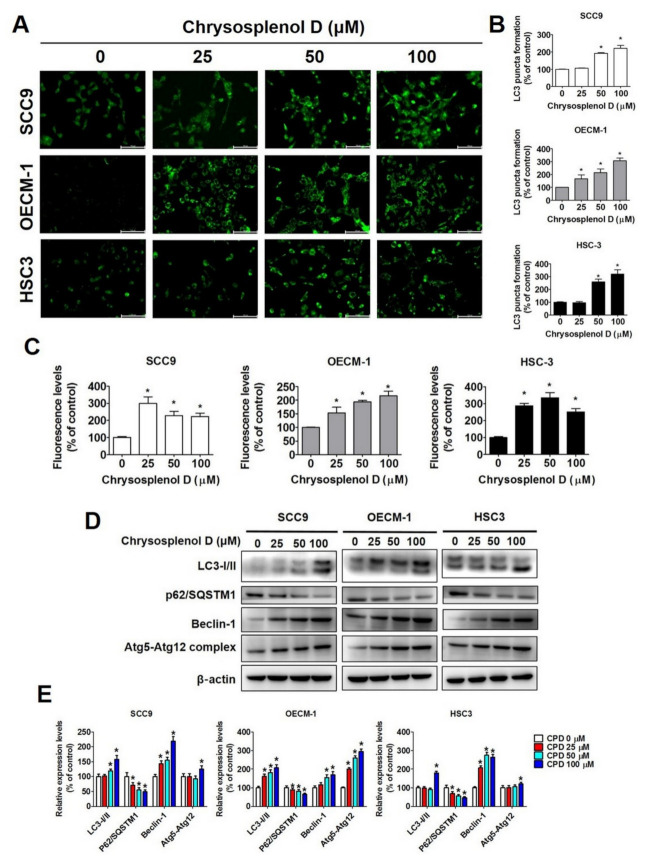
Autophagy induction by chrysosplenol D in OSCC cell lines. (**A**,**B**) SCC-9, OECM-1, and HSC-3 cells were treated with chrysosplenol D for 24 h, followed by immunostaining and the observation of LC-3 under a fluorescence microscope at a magnification of 200×. Graphs show the findings of the statistical analysis of LC-3 formation. (**C**) Chrysosplenol D induced the formation of autophagosomes in OSCC cells that was detected using a fluorescence microplate reader at Ex/Em = 485/530 nm. (**D**,**E**) The protein expression of autophagic markers (LC3-I/II, p62/SQSTM1, Beclin-1, and Atg5–Atg12 complex) was measured through Western blot after chrysosplenol D treatment. The β-actin protein level was used to adjust quantitative results. Graphs show the findings of the statistical analysis of autophagy-related proteins. Data are presented as the mean ± SD from three independent experiments * *p* < 0.05 compared with the vehicle treatment group.

**Figure 7 cancers-13-04327-f007:**
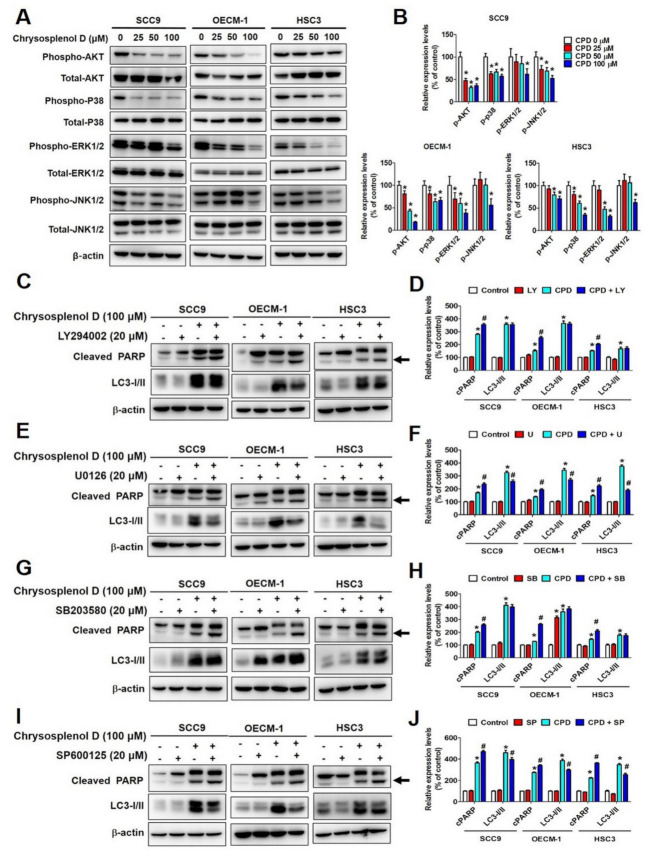
Chrysosplenol D regulated cell apoptosis through the PI3K/AKT and MAPK pathways. (**A**,**B**) After treatment with chrysosplenol D (0, 25, 50, and 100 µM) for 24 h, OSCC cells were collected, and AKT and MAPK pathway proteins were detected using Western blot. The β-actin protein level was used to adjust quantitative results. Graphs show the findings of the statistical analysis of AKT and MAPK proteins. (**C**–**J**) SCC-9, OECM-1, and HSC-3 cells were pretreated with LY294002 (20 μM), U0126 (20 μM), SB203580 (20 μM) or SP600125 (20 μM) for 1 h, followed by chrysosplenol D (0 and 100 μM) for another 24 h. Western blot analysis was performed to detect the protein expression levels of cleaved PARP and LC3-I/II. The β-actin protein level was used to adjust quantitative results. Graphs show the findings of the statistical analysis of cleaved PARP and LC3-I/II proteins. Data are presented as the mean ± SD from three independent experiments * *p* < 0.05 compared with the vehicle treatment group. # *p* < 0.05 compared with chrysosplenol D treatment group.

**Figure 8 cancers-13-04327-f008:**
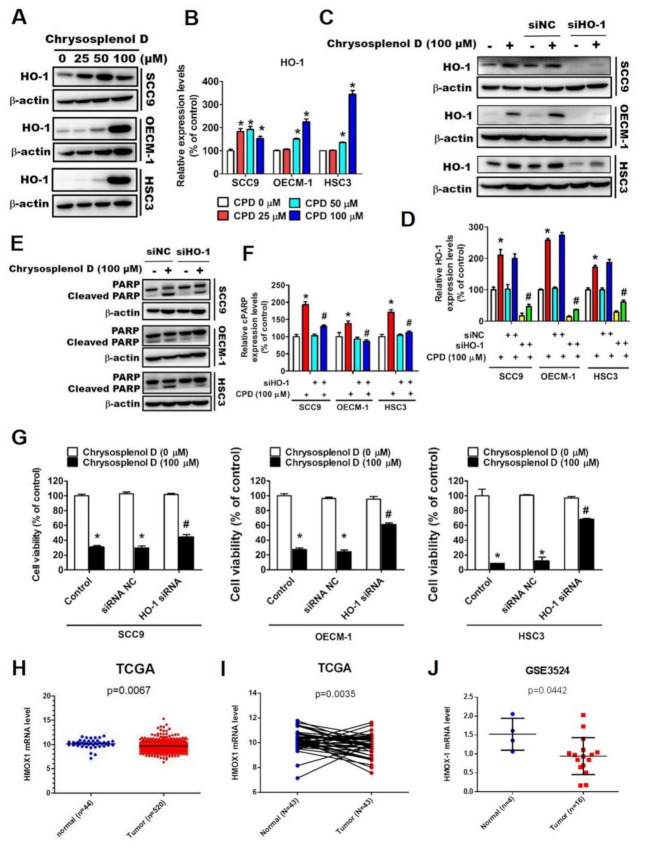
Heme oxygenase-1 plays a crucial rule in chrysosplenol D-induced anti-proliferation and PARP activation. (**A**,**B**) OSCC cells were treated with the indicated doses of chrysosplenol D (0, 25, 50, and 100 μM), and the HO-1 expression level was detected through Western blot analysis. The β-actin protein level was used to adjust quantitative results. Graphs show the findings of the statistical analysis of HO-1 proteins. (**C**–**G**) SCC-9, OECM-1, and HSC-3 cells were transiently transfected with control siRNA or HO-1-specific siRNA and subjected to Western blot analysis and cell viability assay. (**C**,**D**) The knockdown efficiency of HO-1-specific siRNA and the combined effect of HO-1 siRNA and chrysosplenol D were determined by the expression of HO-1. The β-actin protein level was used to adjust quantitative results. Graphs show the findings of the statistical analysis of HO-1 protein. (**E**,**F**) OSCC cells were co-treated with HO-1-specific siRNA and chrysosplenol D, and the PARP expression level was analyzed using Western blot. The β-actin protein level was used to adjust quantitative results. Graphs show the findings of the statistical analysis of PARP protein. (**G**) OSCC cells were co-treated with HO-1-specific siRNA and chrysosplenol D, and cell viability was analyzed using the MTT assay. Data are presented as the mean ± SD from three independent experiments * *p* < 0.05 compared with the vehicle treatment group. # *p* < 0.05 compared with chrysosplenol D treatment group. (**H**) The *HMOX1* mRNA level was analyzed from the head and neck squamous cell carcinoma (HNSCC) dataset, which was retrieved from The Cancer Genome Atlas (TCGA) database, for normal tissues (*n* = 44) and tumor tissues (*n* = 520). (**I**) The *HMOX1* mRNA level in 43 paired cancer tissue samples and normal adjacent tissue samples from the TCGA database. (**J**) The *HMOX1* mRNA expression level of patients with OSCC was analyzed from the Gene Expression Omnibus (GEO) dataset (GSE3524).

## Data Availability

The study did not report any data.

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
