# Peer review of "Chrysosplenol D Triggers Apoptosis through Heme Oxygenase-1 and Mitogen-Activated Protein Kinase Signaling in Oral Squamous Cell Carcinoma"

_cancers, 2021, doi:10.3390/cancers13174327_

Round 1

Reviewer 1 Report

Thank you for this comprehensive and very accurately executed study demonstrating antitumor effects of chrysosplenol D in oral squamous cell carcionomas. This approach is innovative and at the same time a possible starting point for therapeutic approaches.
It would be an important addition if possible approaches in cutaneous squamous cell carcinomas could be presented in addition to the already presented correlations of the HO pathway to oral squamous cell carcinomas. Do the authors see a link between inflammatory stromal responses in squamous cell carcinoma and chrysosplenol D effects ?

Author Response

Thank you for the suggestion. We have noticed that chrysosplenol D suppresses the expression of inflammatory mediator, such as IL-1β, IL-6 and MCP-1, via downregulated IκB and c-JUN phosphorylation in Raw264.7 cells (Flavonoids casticin and chrysosplenol D from Artemisia annua L. inhibit inflammation in vitro and in vivo. Toxicol Appl Pharmacol. 2015 Aug 1;286(3):151-8). Moreover, previous study has pointed the anti-inflammatory effect of HO-1 pathway (Anti-inflammatory actions of the heme oxygenase-1 pathway. Curr Pharm Des. 2003;9(30):2541-51). From above evidences, we can link the anti-inflammatory effect and chrysosplenol D-activated HO-1 pathway. We may further work on the thoroughly investigation on the effects of chrysosplenol D in the future.

Reviewer 2 Report

I congratulate you on the methodological accuracy and the clarity of your presentation, which is easy to understand even for non-experts in the field of molecular biology.

These preclinical results are of great interest also in the light of the search for chemotherapy drugs with a lower systemic toxicity. Future studies should investigate the difference in the molecular effects on tumor lines and healthy cell lines in relation to dose and time of administration.

Author Response

Thank you for the suggestion. As you mentioned, it’s important to find a chemotherapy drug with low systemic toxicity. Our results showed the apoptotic effect and underlying mechanism of chrysosplenol D in OSCC cells. Further study would investigate the appropriate dose and time of chrysosplenol D in normal and tumor cells. We merit this compound may contribute to clinical treatment of oral cancer.